# Electrospinning of Ethylene Vinyl Acetate/Carbon Nanotube Nanocomposite Fibers

**DOI:** 10.3390/polym11030550

**Published:** 2019-03-22

**Authors:** Mária Omastová, Eliška Číková, Matej Mičušík

**Affiliations:** Polymer Institute, Slovak Academy of Sciences, Dúbravská cesta 9, 845 41 Bratislava, Slovakia; matej.micusik@savba.sk (M.M.)

**Keywords:** nanocomposite, electrospinning, ethylene vinyl acetate, carbon nanotubes, XPS, TEM

## Abstract

Nanocomposites, based on an ethylene vinyl acetate (EVA) copolymer with a vinyl acetate content of 34 wt % and varying amounts of multiwall carbon nanotubes (MWCNTs), were prepared by an electrospinning method. The dispersibility of the MWCNTs in the solution was improved by using cholesteryl 1-pyrenecarboxylate (PyChol) as a compatibilizer. The transmission electron microscopy images showed that the MWCNTs were aligned inside of the elastomeric matrix by the electrospinning process. The morphologies of the fibers were evaluated by scanning electron microscopy. When the amount of MWCNTs in the polymer solution reached 3 wt %, fibers with a diameter of 846 ± 447 nm were prepared. The chemical composition of the prepared fibers was investigated by Fourier transform infrared spectroscopy (FTIR) and X-ray photoelectron spectroscopy (XPS). FTIR results confirmed the presence of a carboxyl group, originating from the presence of PyChol. XPS results showed that the EVA fibers produced by electrospinning were oxidized in ethylene units, when comparing the spectra of the original EVA granules, but the presence of MWCNTs enhanced the stability of the EVA. The thermal stabilities of the fibers were tested with thermogravimetric analysis. The results confirmed that the presence of MWCNTs inside the fibers enhanced the thermal stabilities of the prepared nanocomposites.

## 1. Introduction

Ethylene vinyl acetate (EVA), a semicrystalline copolymer, is a flexible and corrosion-resistant material. EVA is an inexpensive and convenient polymer with a variety of properties that highly depend on the ratio of the ethylene and vinyl acetate units [1]. The polyethylene units dominate the structure of EVA copolymers, and they are partially crystalline, while the amorphous vinyl acetate units are flexible, soft and polar. EVA with 10 wt % of VA exhibits lower water absorptivity and higher tensile strength, permeability, Young’s modulus, and dielectric strength values than EVA with 25 wt % or more of vinyl acetate (VAc) [2]. An increase in the VAc content decreases the tensile strength, resistance to heat deformation, and barrier properties of the copolymer. EVA with 33 wt % of VA is used in cable sheaths, hoses, sheeting, ring seals, etc.

Composite materials, based on polymer/carbon nanotube hybrids, are interesting materials because of their enhanced properties, compared to those of the pure polymers. The addition of MWCNTs can enhance a wide range of properties, including the electrical conductivity and the mechanical and thermal properties, of a selected polymeric matrix. MWCNTs are nanosized and have high aspect ratios and low densities. Their dispersion inside a polymeric matrix influences the interactions between the polymer and the MWCNT. The thermal conductivities of EVA composites were investigated by Ghose et al. [3], upon the incorporation of carbon nanotubes, carbon nanofibers, and expanded graphite. Electromechanical actuators and photothermal actuators have been developed, based on ethylene vinyl acetate supplemented with carbon nanotubes [4]. 

Electrospinning has been actively exploited as a versatile and straightforward method for generating ultrathin fibers of various materials, with polymer matrices being the most common [5]. A polyvinylidene difluoride nanofibrous membrane containing MWCNTs and platinum nanoparticles was fabricated by an improved electrospinning technique and tested in biosensor and catalysis applications [6]. Su et al. [7] reported a review about the fabrication of sensors and biosensors on the base of electrospinning of polymers with MWCNTs and and metallic nanoparticles. Tested sensors were successfully applied for sensing gases, such as H_2_S and H_,_ or ethanol, sugar, and H_2_O_2_. 

The parameters affecting the electrospinning process can be divided into three main categories: The solution properties, the ambient parameters, and the variables set by the researcher. The solution properties significant for electrospinning include the viscosity, elasticity, surface tension, and conductivity. The nonwoven polymer-based composite mats that are prepared by electrospinning have been demonstrated to have enhanced properties and applicability and are expected to enable the successful development of new technologies in a wide range of applications [8]. Carbon-based nanofillers, such as carbon black, MWCNTs, single-wall carbon nanotubes (SWCNTs), graphene, and graphene oxide, have been used as fillers in polymeric matrices, and can improve the electrical, thermal, and other properties of the prepared composites. The level of their performance depends significantly on the degree of dispersion and alignment of these carbon fillers [9]. Poly(vinyl acetate) composite nanofibers were prepared by electrospinning, using functionalized and unfunctionalized MWCNTs, showing that functionalized MWCNTs improved dispersion of MWCNTs in composites [10]. Well-dispersed MWCNT in poly(ethylene oxide) (PEO) nucleated polymer crystallization during electrospinning and also acted as a template for polymer orientation [11].

There is a problem with electrospinning EVA with VAc contents lower than 30 wt % under laboratory conditions. EVA solutions need to be heated, e.g., by an infrared lamp, and only by controlling the temperature inside the syringe can EVA be electrospun. EVA (VAc 28 wt %) has previously been electrospun with a filler like clay [12] or clay and iron nanoparticles [13]. Carbon nanotubes have been electrospun in various types of polymers, e.g., polyurethane [14], polycaprolactone [15], polyamide [16], and poly(vinyl alcohol) [17]. However, the electrospinning of EVA with MWCNTs has not been reported. 

In our previous study, photothermal polymeric actuators based on EVA with MWCNTs were prepared by solution casting [4]. Using EVA to create polymeric-based actuating devices was reported for the first time, where it was crucial to achieving a good dispersion of MWCNTs and their alignments. The best actuating results were achieved with 0.3 wt % of MWCNTs in composites, but their production was rather complicated, using a procedure of several steps. The electrospinning technique can produce polymeric composites with aligned MWCNTs, therefore, it initiated our present study. In this work, electrospun nanocomposites based on EVA and MWCNTs were prepared. EVA containing 34 wt % vinyl acetate was used as the polymeric matrix. This copolymer solution remains liquid at laboratory temperatures and is suitable for electrospinning. The amount of MWCNTs was changed from 0.1 to 3.0 wt %. To improve the dispersity and stability of the MWCNTs in the solution, their surface was non-covalently functionalized with a compatibilizer. Fourier transform infrared spectroscopy (FTIR) and X-ray photoelectron spectroscopy (XPS) were used to study the structures and compositions of the prepared composites. The morphologies of the prepared non-woven fibers were investigated by SEM, and the fiber diameters were determined using software. The thermal stabilities of the prepared samples were also investigated.

## 2. Materials and Methods

### 2.1. Materials

A commercially available ethylene vinyl acetate copolymer (EVA, EVA Alcudia PA-461, Repsol, Spain) containing 34 wt % vinyl acetate was electrospun without further purification and used as the polymeric matrix. Multiwall carbon nanotubes (MWCNT, Nanocyl 7000, Nanocyl, Belgium) with a purity of 90%, an outer diameter of 9.5 nm, a length of 1.5 μm, and a surface area of 250–300 m^2^g^−1^ were used as the filler. For the MWCNT surface modification, a non-covalent modifier, cholesteryl 1-pyrenecarboxylate (PyChol), prepared by Polymer Institute, SAS, was used as a surfactant and compatibilizer [4,18]. Chloroform p.a. (CHCl_3_) and ethanol (EtOH, CentralChem, Slovakia) were used as the solvents for sample preparation. Both solvents were used as received, without purification.

### 2.2. Preparation of the Electrospun Fibers

Different types of composite fibers filled with MWCNT concentrations from 0.1 to 3.0 wt % were prepared by electrospinning. The required quantities of the nanofiller and the cholesteryl pyrene carboxylate (PyChol) compatibilizer were dispersed in chloroform and ethanol (70/30 vol %), using a nanofiller/PyChol weight ratio of 1/5, as previously described [18]. Then, the solution was sonicated for 1 h with a Hielscher 400 S sonicator (Hielscher Ultrasonics GmbH, Germany) (~35 μm, ~60 W/cm^2^) at a duty cycle of 100%. After sonication, EVA was added (fixed concentration of 12 wt %), and the final solution was mechanically stirred for 3 h at a speed of 1200 rpm. The solutions were electrospun from a 5 mL syringe with a metallic needle at a flow rate of 1.5 mL/h. A high voltage of 14–15 kV was applied, and the electrospun fibers were collected on aluminium foil. The counter electrode was placed 15 cm from the needle tip.

### 2.3. Characterization of the Prepared Fibers

The morphologies of the electrospun EVA fibers and their average diameters were studied by scanning electron microscopy (SEM) with a JSM Jeol 6610 microscope (Jeol Ltd., Tokyo, Japan) at an accelerating voltage of 10 kV. The samples were sputtered with a thin layer of gold. AzTec software was used to collect the micrographs and process the results. The images were postprocessed using ImageJ software (this program is available at https://imagej.nih.gov/ij/). ImageJ was also used to measure the average diameters of the fibers in the nonwoven fabrics and to estimate the contact angles, as affected by the solvent systems and the concentrations of the electrospun solutions.

A sample was electrospun on a copper TEM grid for 2 s to obtain a single fiber. Transmission electron microscopy observations were performed using a JEOL 1200FX (Jeol Ltd., Tokyo, Japan), operated at an accelerating voltage of 120 kV.

Fourier transform infrared spectroscopy (FTIR) with a NICOLET 8700™ spectrophotometer (Thermo Scientific, Madison, WI, USA), equipped with an ATR (Attenuated Total Reflection) accessory, was used for sample identification. The spectra were measured over the infrared range of 4000–650 cm^−1^. The analyzed area of the samples was approximately 3.14 mm^2^ (the contact area of the Ge crystal, which was used for the ATR measurements), and the average of three measurements taken at different locations was used for each sample. The FTIR spectra were analyzed using OMNIC™ 8.1 software.

For the X-ray photoelectron spectroscopy (XPS) study of the prepared samples, a Thermo Scientific K-Alpha XPS system (Thermo Fisher Scientific, UK), equipped with a micro-focused, monochromatic Al Kα X-ray source (1486.6 eV), was used. The spectra were acquired in constant analyzer energy mode, with a pass energy of 200 eV for the survey scan. Narrow region scans were collected with a pass energy of 50 eV. Charge compensation was achieved with the system flood gun, which provided low energy electrons (approximately 0 eV) and low energy argon ions (20 eV) from a single source. Thermo Scientific Avantage software, version 5.988 (Thermo Fisher Scientific, East Grinstead, UK), was used for digital acquisition and data processing. Spectral calibration was determined by using the automated calibration routine and the internal Au, Ag, and Cu standards supplied with the K-Alpha system. The surface compositions (in atomic %) of the samples were determined by considering the integrated peak areas of the detected atoms and their respective sensitivity factors.

Thermogravimetric analysis (TGA) was carried out under the flow of air (50 cm^3^ min^−1^) at a heating rate of 5 °C min^−1^ with a Mettler Toledo 851e thermogravimetric analyzer (Mettler Toledo GmbH, Switzerland).

## 3. Results

In past work, films based on EVA with MWCNTs were prepared by solution casting to create new types of polymeric photothermal actuators [19]. MWCNTs were used in these studies instead of fillers, such as carbon black or silica, because of their unique properties and high aspect ratio. It is possible to use them at lower filler loading levels than those of other fillers and still achieve the required properties of the actuator. The critical problem of polymeric actuator manufacturing is the dispersion of the MWCNTs into individual nanotubes and their subsequent alignment. The electrospinning method can be a useful tool for creating polymeric materials with the demanding properties described above, as has been reported by other authors [8,10]. While electrospinning the polymer solution containing MWCNTs, the transfer of the shear force from the polymer matrix to the MWCNTs leads to the deagglomeration of the nanotubes and their orientation along the direction of the solution flow. Parameters such as the MWCNT concentration, aspect ratio, and shear rate influence the final fiber properties.

In our previous study [20], the electrospinning of pure EVA with a vinyl acetate content of 28 wt % was very difficult under standard laboratory conditions. Only by continuously heating the polymer solution during the electrospinning process could EVA fibers be produced, but their diameters were in the micrometer range. Subsequently, EVA with slightly higher VAc content of 34 wt % dissolved in chloroform, and ethanol (70/30 vol %) was electrospun under standard laboratory conditions without the application of external heat. The concentrations of the solutions for electrospinning here were chosen as previously described [20]. Chloroform was selected as the solvent because it was previously proven that certain solvents act as selective solvents for the ethylene or VAc units. Ethanol was picked because of its relatively high dielectric constant (ε = 25) and high boiling point (79 °C) compared to those of chloroform (ε = 4.8 and b.p. = 61 °C) [21]. The dielectric constant has been confirmed as one of the key factors of electrospinning in various studies [22,23]. The electrospinning parameters were kept constant, and the properties of the electrospun fibrous mats are discussed with respect to the changing concentration of the nanofiller.

The addition of nanofillers, such as MWCNTs, enhances the conductivity of the electrospinning solution, which leads to thinner fibers as well as improved thermal stabilities and mechanical properties of the prepared nonwoven mats. The basic requirement for the preparation of composite fibers with a nanofiller is good dispersion of the filler within the polymeric matrix. When using MWCNTs, either covalent or non-covalent modification of the filler surface is typically performed to achieve this aim. Non-covalent modification is preferred because it does not disturb the outer CNT layer. A pyrene-based surfactant can be applied and, through π-π interactions with the outer layer of the carbon nanotube, it is able to individualize the MWCNTs. Here, PyChol was applied as the modifier, using a MWCNT/PyChol weight ratio of 1/5, as previously described [18].

Figure 1 shows the color change between electrospun EVA and the composite fibers, with increasing concentrations of MWCNTs. The non-woven mats of pure EVA are white, while those of fibers prepared with 1.0 wt % MWCNTs have visible black spots. When the MWCNT concentration is increased to 3.0 wt %, the collector surface is black, which is further confirmation of MWCNT presence in the prepared fibrous mats. SEM images of pure EVA and EVA with various amounts of nanofiller are shown in Figure 2. 

Figure 3 shows the histograms of the prepared samples of pure EVA and the composites. The average diameter of the pure EVA samples was 2504 ± 982 nm (Figure 3a), due to the low conductivity of the solution. As seen in Figure 3b–f, the average diameters of the electrospun fibers loaded with 3.0 wt % MWCNTs or lower, are all lower than that of pure EVA, varying from 1930 to 1468 nm, with notably high standard deviations (SD). Fibers without beads were produced, which suggests good dispersion of the MWCNTs in the polymer solution. The lowest diameter was measured for the composite fibers that contained 3.0 wt % MWCNTs (Figure 3g). The presence of MWCNTs increased the conductivity of the polymer solution and allowed for better stretching of the electrospinning solution between the needle and target. In the presented images, a few beads were detected in the electrospun fibers. Increasing the concentration of the MWCNTs up to 3.0 wt % resulted in local inhomogeneity in the electrospinning solution and increased the densities of the created fibers, which led to the formation of beads (Figure 2g). However, the increased conductivity of the 3.0 wt % MWCNT solution produced thinner fibers compared to the rest of the prepared samples. EVA/3.0 wt % MWCNT fibers had a diameter of 846 ± 447 nm, as calculated by ImageJ software (Figure 3g). Similar results have been observed in electrospun poly(ε-caprolactone)/carbon nanofiber composites [24]. The fibre diameter prepared by electrospinning decreased with conducting filler content. Decreasing fiber diameter could be attributed to the increased conductivity of the solutions containing a conducting filler as MWCNTs, which increased the likelihood of more electrostatic repulsion, leading to increased stretching, and therefore, thinner fibers produced. This was also documented by other authors, when polyvinylidene fluoride was electrospun with conducting fillers, graphene oxide, and silver particles [25]. 

TEM was used to detect the MWCNTs in the prepared fibers. Figure 4 shows TEM images of the EVA/3.0 wt % MWCNTs. The presence of MWCNT in the fibers is clearly visible. The MWCNTs are encapsulated by the EVA matrix. On a large scale, Figure 4 shows that the MWCNTs have been individualized and are oriented along the axis of the prepared composite fibers. Alignment of MWCNTs in other polymer fibers prepared by electrospinning is also reported by other researchers [6,7]. Both the electric field and mechanical force contributed to MWCNT alignment in electrospun nanofibers. 

The original outer diameter of the MWCNTs was 9.5 nm, and the length was approximately 1.5 μm. Small segments of the MWCNTs are detected, due to breakage of the filler during solution preparation by sonication and high-speed mixing; however, the majority of the nanotube lengths are greater than one micrometer. The average diameter of the MWCNTs obtained from TEM measurements corresponds with that of the original non-treated MWCNTs. 

Figure 5 shows the selected FTIR spectra of EVA, EVA/MWCNT composites, the compatibilizer PyChol, and pure MWCNTs. The FTIR spectra of the electrospun EVA and EVA/MWCNT composite show typical peaks from both the ethylene and vinyl acetate units. The C–H stretching vibrations from ethylene have assigned peaks near 2920 cm^−1^ and 2850 cm^−1^. The typical vibrations of the C=O and C–O in vinyl acetate are visible near 1735 cm^−1^ and 1235 cm^−1^ [26]. In the nanocomposite EVA/MWCNT, there is a unique peak at 1646 cm^−1^. This peak corresponds to the C=O stretching of the carboxyl group in PyChol [27]. The aromatic carbon–carbon stretching vibrations caused by the presence of PyChol are visible in the region of 1500–1625 cm^−1^. The presence of the MWCNTs is evident in the electrospun fiber mats because the color changed from white to almost black, but their presence is not clearly proven by FTIR. Indirect evidence is presented, such as increases in the MWCNT concentration of the fibers, increasing the concentration of the PyChol compatibilizer as well, for which the presence of PyChol was confirmed by FTIR.

Next, XPS was used to characterize the prepared samples. XPS is a surface-sensitive quantitative spectroscopic technique that provides information about the elemental composition and chemical bonding of the surface up to a depth of 10 nm. XPS survey spectra of the pure EVA fibers and all six prepared EVA/MWCNT composites with varying amounts of filler are depicted in Figure 6. More details are provided in Table 1, which summarizes the elemental composition of the sample surfaces. N1s (centered at ~400 eV), C1s (at ~285 eV), O1s (at ~532 eV), Si2p (at ~102 eV), and S2p (at 169 eV) signals are visible. The small amount of detected nitrogen comes from ambient air during the handling, preparation, and collection of the fibers on the aluminium foil. Silica and sulfur are impurities, introduced during processing and sample preparation. In several samples, a small amount of impurities, such as F and Ca, originate from sample preparation and transport for XPS analysis. Pure EVA fibers were analyzed and, because of the possibility of EVA oxidation during electrospinning, for comparison, EVA in the form of the original granules was also studied.

In Figure 7, the C1s and O1s spectra are presented with deconvolution of their peaks. The O1s peak (O1s scan) at approximately 534.7 is attributable to the oxidation of vinyl acetate groups in EVA during the electrospinning process. The oxidation most likely occurs in the methyl group of vinyl acetate, resulting in the surface OH moieties, such as HO*–CO–O [28]. This peak is not present in the rest of the samples, suggesting that the presence of the MWCNTs stabilizes the EVA copolymer during the electrospinning process. Figure 6e also shows that, after the addition of the MWCNTs to the EVA/MWCNT composites, there is a decrease in the signal intensity at about 286 eV, which corresponds to the C-O bond. Table 2 shows the results of the detailed analysis of the C1s and O1s peak fitting for all prepared samples. Comparison of the EVA fibers with the original EVA granules provides evidence of EVA oxidation. Although the O1s peak increase in the electrospun EVA fibers is only 2.8 at % compared to the EVA granules, the ratio of the C=O/C–O bonds significantly decreased when compared with the same ratio in the original EVA granules. This result is similarly repeated when comparing the C–C/C–O ratios from the C1s peak analysis. For the EVA fibers, the C–C/C–O ratio is 67.4/19.3, and in the original EVA granules, it is 87.4/5.4. After the addition of a small amount of the MWCNTs into the EVA matrix, these ratios are closer to those of the original EVA sample. The percentage of C–C in the composite fibers increased with increasing amounts of MWCNTs. Additionally, evidence of the stabilizing effect of the MWCNTs is visible from the C1s peak shape, as depicted in Figure 7, and compared in Figure 7e, where the C1s peak of the electrospun EVA composite fibers containing 0.1, 1.0, and 2.0 wt % MWCNTs are depicted.

Accounting for the quantification of the chemical states in C1s (Table 2) overestimating the overall oxygen content, the oxidation should be connected to the generation of more carbons bonding to oxygen, such as C–O–C or (C=O)–C–O*–C bonds, which would correspond to the shifts of the O1s peak at ca 534.7 eV and the C1s peak at ca 286 eV [28]. Modification of the MWCNTs by the PyChol surfactant was discussed and optimized in [18]. The presence of the MWCNTs in the EVA/MWCNT composites is not possible to prove directly, as it is accompanied only by the less-oxidized carbon (lower C-O intensity at approximately 286 eV, higher C–C intensity at approximately 285 eV).

The thermal stabilities of the prepared composite mats were studied by TGA in air and compared with the stability of the pure EVA fibers. The weight losses in the TGA thermograms of the prepared composite EVA/MWCNT samples, in comparison to that of pure EVA, are shown in Figure 8. Pure MWCNTs have excellent thermal stability [29], and the mass loss for the MWCNTs starts above 500 °C [30]. The difference in the amount of MWCNT filler in the prepared composite fibers was too low to significantly affect the mass loss, and the TGA curves overlap; as such, only data for selected samples are shown. In general, a two-step degradation process is observed for all samples. The TGA thermograms of the EVA show that the first degradation process starts at 270 °C (T_onset_) and is completed at 353 °C (T_f_.). This process corresponds to the loss of acetic acid, as described by Maurin et al. [31]. The second step corresponds to the degradation of the polyethylene chains and starts at approximately 410 °C and ends at 465 °C [32]. These data (Table 3) show that the addition of MWCNTs affected the thermal behavior of the prepared samples. The first degradation step starts at lower temperatures, which can be attributed to the presence of the PyChol stabilizer. The second degradation step is shifted to higher temperatures, as documented by the values of T_max_ and T_f_. Comparing the 10% weight loss for all samples, no significant differences were found. At 50% weight loss, all EVA/MWCNT composite samples are more thermally stable than the pure EVA fibers. The addition of 3.0 wt % MWCNTs increased the temperature of 50% weight loss by approximately 20 °C.

Su at al. [11] reported conductivity of poly(ethylene oxide) containing 3 wt % of /MWCNT electrospun fibers in the range 10^−5^ S/cm, depending on the preparation method. The conductivities of the electrospun EVA/MWCNT mats were also tested by the two-probe method, but since the MWCNTs are coated by the EVA, and their amount is below the percolation threshold, the conductivities were in the range of pure EVA, approximately 10^−9^ S/cm. 

## 4. Conclusions

Electrospun fibrous mat composites of EVA and MWCNTs were successfully prepared. A copolymer of EVA with 34 wt % vinyl acetate was used because it is possible to electrospin its solution at laboratory temperature without additional heating. A homogenous dispersion of the MWCNTs in the polymer solution was obtained, due to the addition of PyChol as a compatibilizer. Composite fibrous mats can be obtained when the concentrations of the MWCNTs in the electrospinning solutions are less than 3 wt %. The presence of MWCNTs within the electrospun fibers was confirmed by TEM analysis. Using analytical methods, such as XPS and FTIR, the presence of the PyChol compatibilizer in the prepared non-woven mats was verified. XPS analysis further showed that the presence of the MWCNTs enhanced the stability of the EVA copolymer, which undergoes oxidation of the ethylene units. The thermal stability of the EVA/MWCNT composites increased in comparison to the pure EVA fibers, as confirmed by TGA measurements.

## Figures and Tables

**Figure 1 polymers-11-00550-f001:**
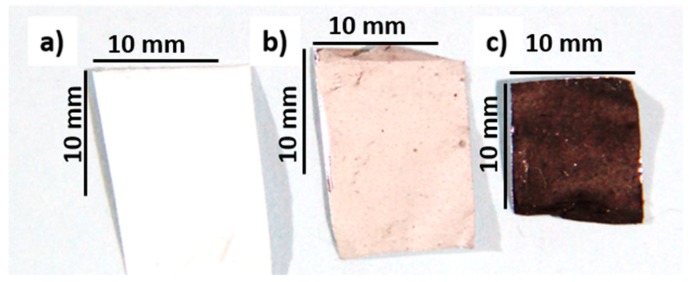
Images of selected electrospun samples collected on aluminium foil: (**a**) Pure ethylene vinyl acetate (EVA), (**b**) EVA/1 wt % multiwall carbon nanotubes (MWCNTs), and (**c**) EVA/3 wt % MWCNTs.

**Figure 2 polymers-11-00550-f002:**
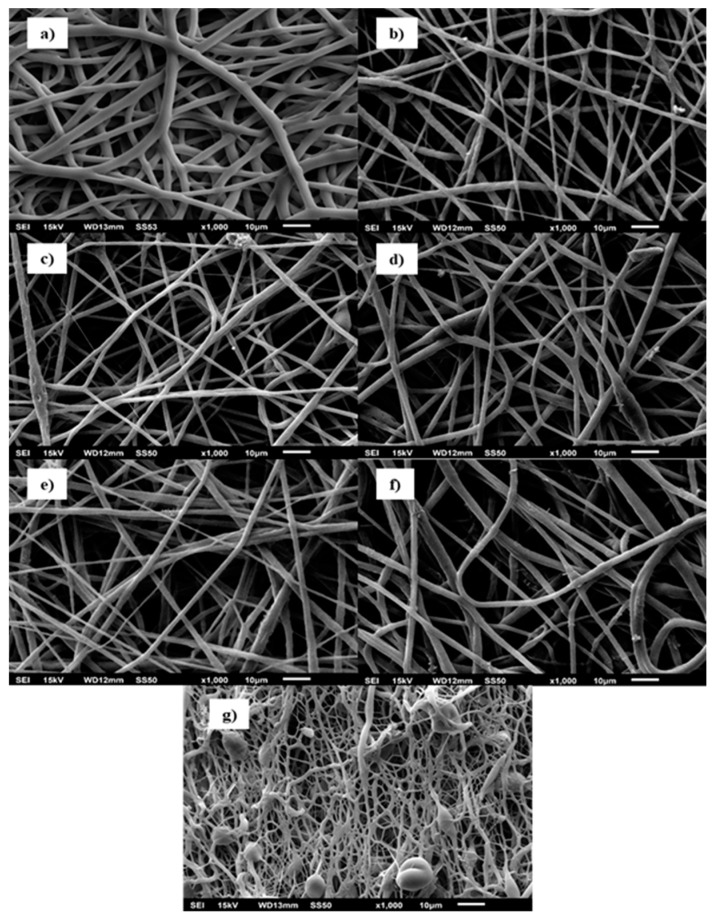
SEM images of (**a**) pure EVA, (**b**) EVA/0.1 wt % MWCNTs, (**c**) EVA/1.0 wt % MWCNTs, (**d**) EVA/1.5 wt % MWCNTs, (**e**) EVA/2.0 wt % MWCNTs, (**f**) EVA/2.5 wt % MWCNTs, and (**g**) EVA/3.0 wt % MWCNTs.

**Figure 3 polymers-11-00550-f003:**
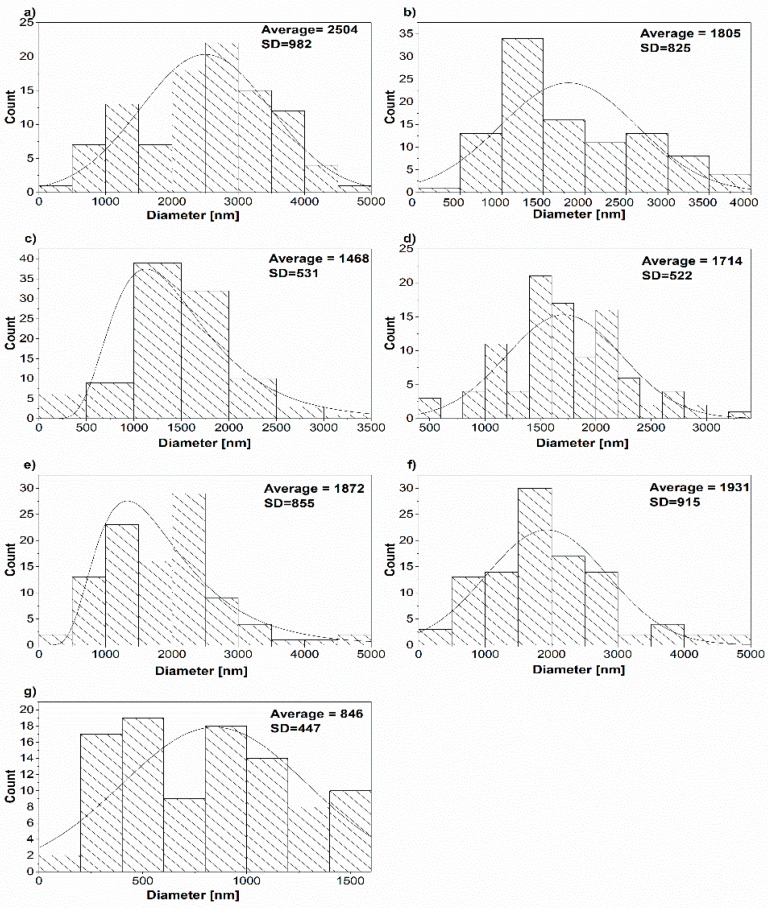
Histograms of the prepared samples: (**a**) Pure EVA, (**b**) EVA/0.1 wt % MWCNTs, (**c**) EVA/1.0 wt % MWCNTs, (**d**) EVA/1.5 wt % MWCNTs, (**e**) EVA/2.0 wt % MWCNTs, (**f**) EVA/2.5 wt % MWCNTs, and (**g**) EVA/3.0 wt % MWCNTs.

**Figure 4 polymers-11-00550-f004:**
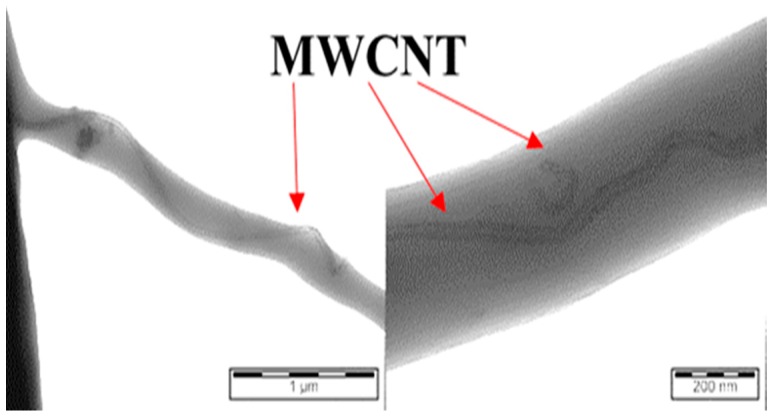
TEM images of EVA/3.0 wt % MWCNT.

**Figure 5 polymers-11-00550-f005:**
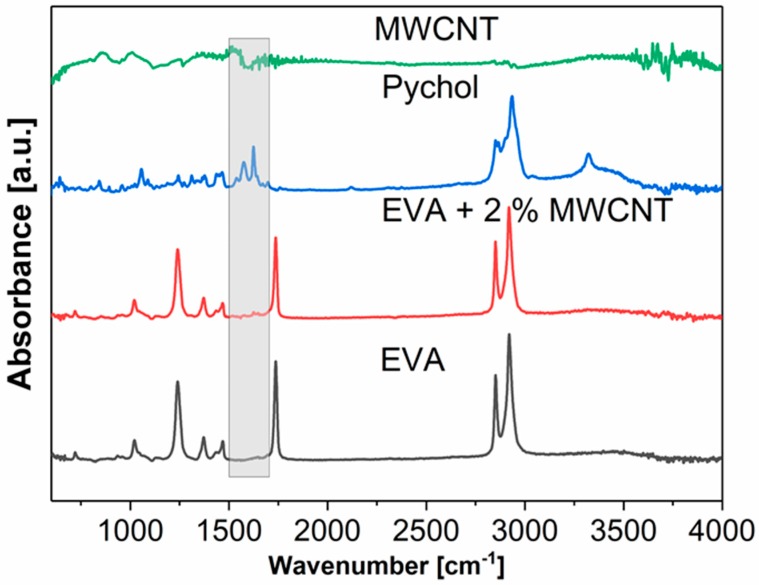
Comparison of the Fourier transform infrared spectroscopy (FTIR) spectra of EVA, the compatibilizer cholesteryl 1-pyrenecarboxylate (PyChol), MWCNTs, and EVA/2.0 wt % MWCNTs.

**Figure 6 polymers-11-00550-f006:**
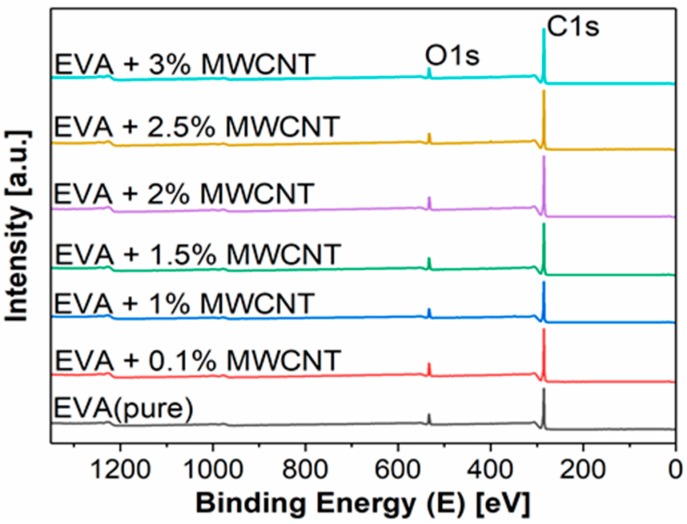
X-ray photoelectron spectroscopy (XPS) survey spectra of prepared samples of pure EVA fibers and EVA/MWCNT composites with various wt % of filler.

**Figure 7 polymers-11-00550-f007:**
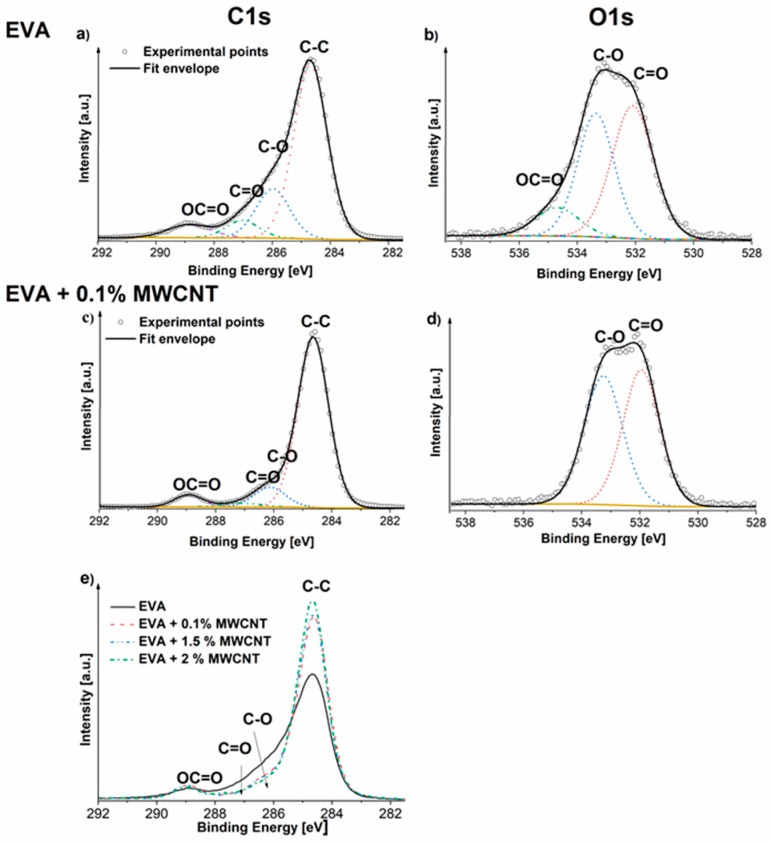
C1s and O1s scans of electrospun (**a**) and (**b**) EVA, (**c**), and (**d**) EVA/ 0.1 wt % MWCNTs, and (**e**) C1s comparison of selected samples.

**Figure 8 polymers-11-00550-f008:**
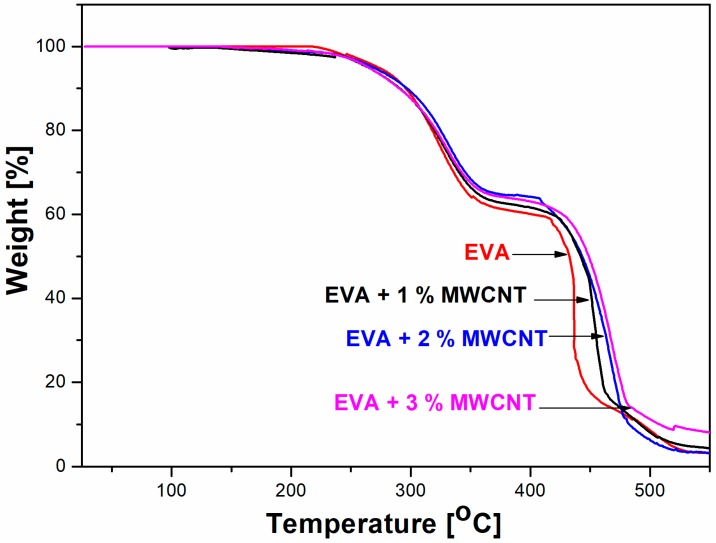
Thermogravimetric analysis (TGA) curves of selected samples.

**Table 1 polymers-11-00550-t001:** Apparent surface chemical composition of EVA matrix, MWCNTs and surface modified MWCNTs, and EVA/MWCNT fibers, as determined by XPS.

Sample	Surface Chemical Composition [at %]					
	C1s	O1s	N1s	Si2p	S2p	F1s/Ca2p
EVA (granules)	87.2	7.7	3.1	2.0	-	-/-
EVA fibers	89.2	10.5	0.2	0.1	-	-/-
MWCNT	99.5	0.5	-	-	-	-/-
PyChol	93.5	4.3	1.7	0.5	-	-/-
PyChol + MWCNTs	89.6	7.8	2.5	-	-	0.2/-
EVA + 0.1 wt % MWCNTs	89.0	10.7	-	0.3	-	-/-
EVA + 1.0 wt % MWCNTs	88.2	10.1	0.4	0.4	0.3	-/0.5
EVA + 1.5 wt % MWCNTs	89.3	10.2	0.4	0.1	-	-/-
EVA + 2.0 wt % MWCNTs	88.6	9.7	1.1	0.4	0.1	-/0.2
EVA + 2.5 wt % MWCNTs	89.0	8.4	1.7	0.9	-	-/-
EVA + 3.0 wt % MWCNTs	89.7	8.8	1.2	0.4	-	-/-

**Table 2 polymers-11-00550-t002:** Elemental surface composition (in at %).

Sample	Surface Chemical Composition [at %]	
	C1s C–C//C–O//C=O/OC=O (284.7//286.0//287.3/288.9)	O1s C=O/C–O/HO*–CO–O (532.1/533.4/534.7)
EVA (granules)	87.3 87.4//5.4//5.5//1.7	7.7 78.4/21.6/-
EVA	89.2 67.4//19.3//6.7/6.6	10.5 48.8/40.2/11.1
EVA/0.1 wt % MWCNT	89.0 82.1//9.9//2.5/5.5	10.7 50.2/49.8/-
EVA/1.0 wt % MWCNT	88.2 85.9//6.5//2.9/4.7	10.1 65.0/35.0/-
EVA/1.5 wt % MWCNT	89.3 83.3//8.3//3.0/5.4	10.2 50.7/49.3/-
EVA/2.0 wt % MWCNT	88.6 86.2//7.6//1.9/4.3	9.7 63.3/36.7/-
EVA/2.5 wt % MWCNT	89.0 86.3//7.0//2.9/3.8	8.4 65.7/34.3/-
EVA/3.0 wt % MWCNT	89.7 85.9//7.7//2.1/4.3	8.8 63.7/36.3/-

**Table 3 polymers-11-00550-t003:** Degradation temperatures at 10 and 50% mass loss for selected samples.

Sample	T_10%_ [°C]	T_50%_ [°C]	First Degradation Step (T_d1_)			Second Degradation Step (T_d2_)		
			T_onset_ [°C]	T_max_ [°C]	T_f_ [°C]	T_onset_ [°C]	T_max_ [°C]	T_f_ [°C]
EVA	301	433	270	323	353	410	436	465
EVA/1.0 wt % MWCNT	302	440	251	331	374	414	453	469
EVA/2.0 wt % MWCNT	303	443	222	330	369	416	467	483
EVA/3.0 wt % MWCNT	303	451	252	328	374	425	467	487

Peak temperatures: T_onset_ = process start, T_max_ = peak maximum, and T_f_ = process finish.

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
