# Peer review of "Electrospinning of Ethylene Vinyl Acetate/Carbon Nanotube Nanocomposite Fibers"

_polymers, 2019, doi:10.3390/polym11030550_

Round 1
Reviewer 1 Report
In this manuscript, the authors reported the fabrication of EVA/CNTs nanocomposite fibers via electrospinning. The fabricated EVA/CNTs fibers were characterized with multi-techniques and the thermal property of fibers were also tested. It can be found that the structure and property of electrospun EVA/CNTs fibers were related to the contents of used CNTs. Previously, a lot of studies on the electrospun synthesis of polymer/CNTs hybrid fibers/nanofibers have been reported, and the effects of CNTs to the structure and properties of fibers were obtained. In this point, the novelty and significance of this work are not good for the publication. However, as the authors performed detailed studies on EVA/CNTs fibers in this work, I can recommend this manuscript to publish at Polymers after major revisions.
Special comments:
The part of “Abstract” should be modified. The authors provided only some general information in this part, however some important results are not given.
In the “Introduction” part, it is suggested for the authors to add a few more sentences to make it more clear for the novelty and significance of their work. In addition, several very relevant refs (such as Nanotechnology, 2006, 17, 5829; RSC Adv, 2014, 4, 52598; Carbon, 2012, 50, 5605; ACS AMI, 2014, 6, 7563.) are suggested to add.
In Figure 1, it is better to give a scale bar for the images.
In Figure 2g, the authors found that 3.0% CNTs could greatly decrease the diameter of fabricated EVA/CNTs fibers. Why? The authors should give more discussion on that.
For Figure 4, it is suggested for the authors to add more discussion on why the CNTs can be oriented along the electrospun polymer fibers.
The conductivity and mechanical properties of the fabricated EVA/CNTs fibers should be performed. It is suggested for the authors to refer to the Carbon (2012, 50, 5605) paper.
Author Response
The part of “Abstract” should be modified. The authors provided only some general information in this part, however some important results are not given.
Abstract was extended and modified, and the insertions are in MS marked in color.
Abstract: Nanocomposites based on an ethylene vinyl acetate (EVA) copolymer with a vinyl acetate content of 34 wt. % and varying amounts of multiwall carbon nanotubes (MWCNT) were prepared by an electrospinning method. The dispersibility of the MWCNT in solution was improved by using cholesteryl 1-pyrenecarboxylate (PyChol) as a compatibilizer. The transmission electron microscopy images showed that the MWCNT were aligned inside of the elastomeric matrix by the electrospinning process. The morphologies of the fibers were evaluated by scanning electron microscopy. When the amount of MWCNT in polymer solution reached 3.0 wt.% fibers with diameter 846 ± 447 nm were prepared. The chemical compositions of the prepared fibers were investigated by Fourier transform infrared spectroscopy (FTIR) and X-ray photoelectron spectroscopy (XPS). FTIR results confirmed presence of carboxyl group originated from PyChol presence. XPS results showed that the EVA fibers produced by electrospinning are oxidized in ethylene units, when compare the spectra of original EVA granules, but presence of MWCNT enhanced the stability of the EVA. The thermal stabilities of the fibers were tested with thermogravimetric analysis. The results confirmed that the presence of MWCNT inside of the fibers enhanced the thermal stabilities of the prepared nanocomposites.
In the “Introduction” part, it is suggested for the authors to add a few more sentences to make it more clear for the novelty and significance of their work. In addition, several very relevant refs (such as Nanotechnology, 2006, 17, 5829; RSC Adv, 2014, 4, 52598; Carbon, 2012, 50, 5605; ACS AMI, 2014, 6, 7563.) are suggested to add.
Explanation of the study was added, and we hope also novelty is not disputable. EVA/MWCNT electrospun fibers is reported first time.
In our previous study photothermal polymeric actuators based on EVA with MWCNT were prepared by solution casting. It was first time reported using EVA for creating polymeric based actuating devices, where crucial was achieving good dispersion of MWCNT and their alignments. The best actuating results were achieved with 0.3 wt% of MWCNT in composites, but their production was rather complicated using few steps procedure. Electrospinning technique can produce polymeric composites with aligned MWCNT, therefore it was initiating the present study
Recommended papers were added to the Introduction, now ref. 6,7, 10 and 11.
In Figure 1, it is better to give a scale bar for the images.
Scale bare was added to Fig. 1.
In Figure 2g, the authors found that 3.0 % CNTs could greatly decrease the diameter of fabricated EVA/CNTs fibers. Why? The authors should give more discussion on that.
New paragraph was added.
The fibre diameter prepared by electrospinning decreasing with conducting filler content. Decreasing fibre diameter could be attributed to the increased conductivity of the solutions containing conducting filler as MWCNT, which increased the likelihood of more electrostatic repulsion leading to increased stretching, therefore thinner fibres are produced. It was also documented by other authors, when polyvinylidene fluoride was electrospun with conducting fillers, graphene oxide and silver particles [25].
New reference was added, now ref. 25.
For Figure 4, it is suggested for the authors to add more discussion on why the CNTs can be oriented along the electrospun polymer fibers.
Discussion was appended.
Alignment of MWCNTs in other polymers fibers prepared by electrospinning is reported also by other researchers [6, 7]. Both the electric field and mechanical force contributed to MWCNT alignment in electrospun nanofibers.
The conductivity and mechanical properties of the fabricated EVA/CNTs fibers should be performed. It is suggested for the authors to refer to the Carbon (2012, 50, 5605) paper.
In the last part of Results and discussion we mentioned conductivity testing, following paragraph was added.
Su at al. [11] reported conductivity of poly(ethylene oxide) containing 3 wt.% of /MWCNT electrospun fibers in the range 10−5 S/cm, depending on preparation method. The conductivities of the electrospun mats were also tested by the two probe method, but since the MWCNT are coated by the EVA, and their amount is below percolation threshold, the conductivities were in the range of pure EVA, approximately 10−9 S/cm.
Mechanical properties were not tested, we suppose that paper is valuable also without this particular testing. We have not now enough electrospun material for performing these test.
Reviewer 2 Report
The reviewed work does not provide any valuable information beyond the fact that the authors did it and that they performed some characterization of the electrospun EVA/CNT composite mats.
It looks that, except for the color change, no other modification of EVA characteristics have been observed. It is also not clear what was the purpose of the whole work. If the goal was to just spin and do some characterization, that’s good for the supplementary information but it is not enough for a scientific paper.
Figure 4 is valuable, but was it just the only one fiber fragment, where alignment of the nanotube was observed or most of the nanotubes were extended and aligned in the direction of the nanofibers? The fiber diameter of the fibers shown in Figure 4 is less than 1 micron but most of the fibers were of much greater diameter, so what was the degree of extension/alignment of CNTs in those bigger diameter fibers?
If generation of fibers with extended/aligned CNT was the major focus of the paper, the authors should provide a more detailed analysis of why it would be important and how that changed with elctrospinning conditions or CNT loading in the fibers. Alternatively, the authors should show that the addition of CNT to the EVA fibers changes something more than by a negligibly small fraction. For example, what about mechanical characteristics?
Author Response
Comments and Suggestions for Authors
The reviewed work does not provide any valuable information beyond the fact that the authors did it and that they performed some characterization of the electrospun EVA/CNT composite mats.
It looks that, except for the color change, no other modification of EVA characteristics have been observed. It is also not clear what was the purpose of the whole work. If the goal was to just spin and do some characterization, that’s good for the supplementary information but it is not enough for a scientific paper.
We would like to emphasis, that the electrospinning of EVA with MWCNT has not been reported before, so it is first time we have done it. A copolymer of EVA with 34 wt.% vinyl acetate was used because it is possible to electrospun its solution at laboratory temperature without additional heating.
The introduction part was extended giving more reason for this study.
In our previous study photothermal polymeric actuators based on EVA with MWCNT were prepared by solution casting. It was first time reported using EVA for creating polymeric based actuating devices, where crucial was achieving good dispersion of MWCNT and their alignments. The best actuating results were achieved with 0.3 wt% of MWCNT in composites, but their production was rather complicated using few steps procedure. Electrospinning technique can produce polymeric composites with aligned MWCNT, therefore it was aim our present study.
Figure 4 is valuable, but was it just the only one fiber fragment, where alignment of the nanotube was observed or most of the nanotubes were extended and aligned in the direction of the nanofibers? The fiber diameter of the fibers shown in Figure 4 is less than 1 micron but most of the fibers were of much greater diameter, so what was the degree of extension/alignment of CNTs in those bigger diameter fibers?
For TEM analysis a sample was electrospun on a copper TEM grid for 2 s only to obtain a single fibers. This fact probably distort results discussed in Fig 4, but it was only possibility how to perform TEM study. We investigated more fibers, where we found similar MWCNT alignment in EVA fibre.
If generation of fibers with extended/aligned CNT was the major focus of the paper, the authors should provide a more detailed analysis of why it would be important and how that changed with elctrospinning conditions or CNT loading in the fibers. Alternatively, the authors should show that the addition of CNT to the EVA fibers changes something more than by a negligibly small fraction. For example, what about mechanical characteristics?
In the introduction part we added new paragraph, and also comment about conductivity study was added in Results and discussion.
Mechanical properties were not tested, we have not now enough amount of composites for this testing. We suppose that very important information is results of XPS study.
XPS analysis confirm oxidation of EVA copolymer during electrospinning process, which undergoes of the ethylene units, as shown by comparison of original EVA in the form of pellets and pure EVA fibres. XPS further showed that the presence of the MWCNT enhanced the stability of the EVA copolymer, what is also very important fact for composite production.
Round 2
Reviewer 1 Report
In this revised version, the authors made great improvements according to my comments and suggestions. I am satisfied with these changes, and therefore recommend the publication of this manuscript at Polymers with the current version.
Reviewer 2 Report
This is a revised manuscript. The authors did a great job in improving the previous version of their writing. The manuscript is now suitable for publication.